# Efficiently Computing Transitions in Cartesian Abstractions

**Primary Keywords:** *None*

## Abstract

Counterexample-guided Cartesian abstraction refinement yields strong heuristics for optimal classical planning. The approach iteratively finds a new abstract solution, checks where it fails for the original task and refines the abstraction to avoid the same failure in subsequent iterations. The main bottleneck of this refinement loop is the memory needed for storing all abstract transitions. To address this issue, we introduce an algorithm that efficiently computes abstract transitions on demand. This drastically reduces the memory consumption and allows us to solve tasks during the refinement loop and during the search that were previously out of reach.

## Introduction

The most common approach to solving classical planning tasks optimally is $A^*$ search (Hart, Nilsson, and Raphael 1968) with an admissible heuristic (e.g., Helmert and Domshlak 2009; Karpas and Domshlak 2009; Katz and Domshlak 2010; Pommerening et al. 2015; Sievers and Helmert 2021). Heuristics based on abstractions of the planning task have been particularly successful. Examples for abstraction-based heuristics are, in order of increasing generality: pattern databases (PDBs; Edelkamp 2001; Sievers, Ortlieb, and Helmert 2012), domain abstractions (Kreft et al. 2023), and Cartesian abstractions (Seipp and Helmert 2018).

A prominent way of generating such abstractions is counterexample-guided abstraction refinement (CEGAR) (Clarke et al. 2003). Since the introduction of CEGAR for classical planning in the context of Cartesian abstractions (Seipp and Helmert 2013), the method has also been adapted to PDBs (Rovner, Sievers, and Helmert 2019) and domain abstractions (Kreft et al. 2023). Furthermore, CEGAR has been used to create PDBs and Cartesian abstractions for probabilistic planning tasks (Klößner et al. 2022; Klößner, Seipp, and Steinmetz 2023).

CEGAR starts with a very coarse abstraction and then iteratively finds a cheapest abstract solution, checks where it fails for the original task and refines the abstraction to avoid the same flaw in subsequent iterations by splitting the state that caused the flaw into two new states. If CEGAR finds a solution for the original task during the refinement process, it is guaranteed to be optimal. Otherwise, the resulting abstraction can be used as a heuristic for an $A^*$ search.

Since domain abstractions and especially PDBs do not allow for fine-grained refinement, it is infeasible to solve nontrivial tasks while refining these types of abstractions. Therefore, existing approaches for these abstraction types mainly create *collections* of abstractions focusing on different aspects of the task (e.g., Haslum et al. 2007; Pommerening, Röger, and Helmert 2013; Franco et al. 2017; Seipp 2019).

Cartesian abstractions, however, allow for fine-grained refinements since each iteration only adds one additional state. Consequently, Cartesian CEGAR is able to solve large tasks during the refinement loop. Previously, the main bottlenecks of the refinement loop in the classical planning setting were the times for finding the next cheapest solution and the next flaw in it, but these two bottlenecks have been addressed recently by *incrementally* revising all cheapest paths (Seipp, von Allmen, and Helmert 2020) and by finding and addressing *batches* of flaws (Speck and Seipp 2022).

Now, the main bottleneck of the refinement loop is the *memory* needed for storing the abstract transitions. It is well known that storing abstract transitions, not abstract states, is the limiting factor for abstractions. In Cartesian abstractions, the problem is especially severe since we need access to both the incoming and outgoing transitions of a state in order to efficiently rewire the transition system after a refinement step. Merge-and-shrink abstractions address the problem by using *label reduction* (Sievers and Helmert 2021). PDBs and domain abstractions circumvent the issue by computing abstract transitions on demand (Rovner, Sievers, and Helmert 2019; Kreft et al. 2023). To do this efficiently, they use perfect hashing (Sievers, Ortlieb, and Helmert 2012) and the *successor generator* data structure (Helmert 2006).

Cartesian abstractions are too general to allow for perfect hashing. However, they are specific enough that a successor generator can efficiently compute the operators $o$ applicable in abstract state $a$. To efficiently compute which abstract states $b$ can be reached from $a$ by applying $o$, we turn to the abstraction's *refinement hierarchy*, which records all splits during the refinement loop in a tree data structure.

In our experiments, we show that computing transitions on demand drastically reduces the memory footprint and thus increases the number of tasks solved during the refinement loop. For the remaining tasks, we obtain much better heuristic estimates than before and consequently solve many additional tasks in the ensuing $A^*$ search.

## Background

A SAS$^+$ planning task (Bäckström and Nebel 1995) is a tuple $\Pi = \langle \mathcal{V}, \mathcal{O}, s_0, s_\star \rangle$, where $\mathcal{V} = \langle v_1, \ldots, v_n \rangle$ is a finite sequence of state variables, each with an associated finite domain $dom(v_i)$. An *atom* is a pair $\langle v, d \rangle$ with $v \in \mathcal{V}$ and $d \in dom(v)$. A *partial state* $s$ maps a subset $\mathcal{V}(s)$ of $\mathcal{V}$ to values $s[v] \in dom(v)$ for $v \in \mathcal{V}(s)$. If $\mathcal{V}(s) = \mathcal{V}$, we call $s$ a *state*. The set of all states in $\Pi$ is $S(\Pi)$. We often treat partial states as sets of atoms. *Updating* partial state $p$ with partial state $q$ results in partial state $r = p \oplus q$, with $r[v] = q[v]$ for all $v \in \mathcal{V}(q)$, and $r[v] = p[v]$ for all $v \in \mathcal{V}(p) \setminus \mathcal{V}(q)$.

Each operator $o \in \mathcal{O}$ is a pair $\langle pre(o), post(o) \rangle$, where $pre(o)$ and $post(o)$ are partial states specifying the *precondition* and *effect* of $o$. The *postcondition* of $o$ is $post(o) = pre(o) \oplus eff(o)$. Operator $o$ is applicable in state $s$ if $pre(o) \subseteq s$ and applying $o$ in $s$ results in state $s[\![o]\!] = s \oplus eff(o)$. The cost of $o$ is $cost(o) \in \mathcal{R}_0^+$. The *initial state* $s_0$ is a state and the *goal* $s_\star$ is a partial state. Solving $\Pi$ optimally implies finding a cheapest iteratively-applicable sequence of operators that transforms $s_0$ into a state $s$ with $s_\star \subseteq s$.

A task $\Pi$ induces a *transition system* $\mathcal{T}$ which is a directed, labeled graph with states $S(\mathcal{T}) = S(\Pi)$, labels $L(\mathcal{T}) = \mathcal{O}$, transitions $T(\mathcal{T}) = \{s \xrightarrow{o} s[\![o]\!] \mid o \in \mathcal{O}, s \in S(\mathcal{T}), pre(o) \subseteq s\}$, initial state $s_0(\mathcal{T}) = s_0$ and goal states $S_\star(\mathcal{T}) = \{s \mid s \in S(\mathcal{T}), s_\star \subseteq s\}$.

An *abstraction* $\sim$ of $\mathcal{T}$ is an equivalence relation over $S(\mathcal{T})$ (Seipp and Helmert 2018). It induces an *abstract* transition system $\mathcal{T}'$ with states $S(\mathcal{T}') = \{[s]_\sim \mid s \in S(\mathcal{T})\}$, labels $L(\mathcal{T}') = L(\mathcal{T})$, transitions $T(\mathcal{T}) = \{[s]_\sim \xrightarrow{o} [s']_\sim \mid s \xrightarrow{o} s' \in T(\mathcal{T})\}$, initial state $[s_0]_\sim$ and goal states $\{[s]_\sim \mid s \in S_\star(\mathcal{T})\}$. An abstract state is *Cartesian* if it has the form $A_1 \times \ldots \times A_n$, where $A_i = dom(v_i, a) \subseteq dom(v_i)$ for all $1 \leq i \leq |\mathcal{V}|$. An abstraction is Cartesian if all its states are Cartesian. A partial state $p$ induces the Cartesian set $\mathcal{C}(p) = A_1 \times \ldots \times A_n$, with $A_i = \{p[v_i]\}$ if $v_i \in \mathcal{V}(p)$ and $A_i = dom(v_i)$ otherwise.

The intersection of two Cartesian sets $a = A_1 \times \ldots \times A_n$ and $b = B_1 \times \ldots \times B_n$ is $a \cap b = (A_1 \cap B_1) \times \ldots \times (A_n \cap B_n)$. The *regression* of Cartesian set $b = B_1 \times \ldots \times B_n$ over operator $o \in \mathcal{O}$ is $regr(b, o) = A_1 \times \ldots \times A_n$ with

$$A_i = \begin{cases} B_i & \text{if } v_i \notin \mathcal{V}(post(o)) \\ \emptyset & \text{if } v_i \in \mathcal{V}(post(o)) \text{ and } post(o)[v_i] \notin B_i \\ pre(o)[v_i] & \text{if } v_i \in \mathcal{V}(pre(o)) \text{ and } post(o)[v_i] \in B_i \\ dom(v_i) & \text{otherwise.} \end{cases}$$

Similarly, the *progression* of Cartesian set $a = A_1 \times \ldots \times A_n$ over operator $o \in \mathcal{O}$ is $progr(a, o) = B_1 \times \ldots \times B_n$ with

$$B_i = \begin{cases} A_i & \text{if } v_i \notin \mathcal{V}(post(o)) \\ \emptyset & \text{if } v_i \in \mathcal{V}(pre(o)) \text{ and } pre(o)[v_i] \notin A_i \\ post(o)[v_i] & \text{otherwise.} \end{cases}$$

## Efficiently Computing Transitions

For Cartesian CEGAR, we need to efficiently obtain both the incoming and outgoing transitions of a given abstract state.[1] Traditionally, this has been done by storing all transitions explicitly and rewiring them after each refinement step. Since this approach often quickly consumes huge amounts of memory, we now present an approach that computes the transitions on demand.

We begin by defining under which conditions a Cartesian abstraction contains a given transition.

**Proposition 1.** *Cartesian abstraction $\mathcal{T}'$ contains a transition from state $a \in S(\mathcal{T}')$ to state $b \in S(\mathcal{T}')$ via operator $o \in \mathcal{O}$ iff $a \cap \mathcal{C}(pre(o)) \neq \emptyset$ and $b \cap progr(a, o) \neq \emptyset$.*

*Proof sketch.* Seipp and Helmert (2018) define the function CHECKTRANSITION($a$, $o$, $b$) for this computation (see their Algorithm 5). It returns true iff three conditions are met: 1) all precondition atoms $\langle v, d \rangle$ are part of $dom(a, v)$, 2) all postcondition atoms $\langle v, d \rangle$ are part of $dom(b, v)$ and 3) for all variables $v$ not mentioned by the operator the two abstract domains $dom(a, v)$ and $dom(b, v)$ intersect. We can distill the three conditions into two tests that check whether two Cartesian sets intersect. From condition 1) we get $a \cap \mathcal{C}(pre(o)) \neq \emptyset$, and from conditions 2) and 3) we get $b \cap progr(a, o) \neq \emptyset$. $\square$

The formulation of the transition check from Proposition 1 reveals that we can divide the task of computing the outgoing transitions of an abstract state $a$ into two steps: first we compute the set of operators $o$ that are applicable in $a$, then we compute the set of abstract states $b$ reachable from $a$ via transitions labeled with $o$. We use the same two-step approach for computing incoming transitions.

## Outgoing Operators

The set $\mathcal{O}_{out}(a)$ of operators applicable in abstract state $a$ is $\{o \in \mathcal{O} \mid \mathcal{C}(pre(o)) \cap a \neq \emptyset\}$. The naive way of computing $\mathcal{O}_{out}(a)$ is to iterate over $\mathcal{O}$ and checking each operator for applicability. We can improve over this computation by exploiting the fact that $a$ is a Cartesian set, which allows us to feed it into the *successor generator* data structure, developed for efficiently enumerating all operators applicable in a given *concrete* state (Helmert 2006).

In its original form, the successor generator is a tree data structure, where each internal node $n$ branches over a subset of the values $d \in dom(n.var)$ of a variable $n.var$. Additionally, internal nodes have a child node for the "don't care" value $\top$. When querying a successor generator for a given concrete state $s$, at each node $n$ we follow the "don't care" child and the child for $s[n.var]$ if it is defined. The traversal stops at the leaf nodes, which store the sets of applicable operators. A successor generator only needs space $O(\sum_{o \in \mathcal{O}} |pre(o)| + eff(o)|)$ and querying it is usually sublinear in the number of operators, and in the best case only linear in the number of *applicable* operators (Sievers, Ortlieb, and Helmert 2012).

Since our abstract states are Cartesian, we can reuse the successor generator data structure with minimal adaptation.

---

[1] While a plain forward search would only need outgoing transitions, it is much faster to find cheapest paths with incremental search, which requires access to both incoming and outgoing transitions (Seipp, von Allmen, and Helmert 2020).

**Algorithm 1** Compute all applicable operators for a given abstract state $a$. The recursive algorithm is called with $n$ set to the root node of the successor generator.

1: **function** $\mathcal{O}_{out}(a, n)$
2:    **if** $n$ is leaf **then**
3:       **yield** $n.operators$
4:    **else**
5:       **for each** child $\in n.children$ **do**
6:          **if** $child.val \in \{\top\} \cup dom(a, n.var)$ **then**
7:             **yield from** $\mathcal{O}_{out}(a, child)$

We can construct the successor generator in exactly the same way as for concrete states. Only the querying needs to be altered: instead of testing only the single value $s[n.var]$ in each internal node $n$, we now follow all child nodes whose value $d \in dom(n.var)$ is contained in the abstract domain $dom(a, n.var)$. Algorithm 1 shows pseudo-code.

**Proposition 2.** *Given an abstract state $a$ and the root node $n$ of a successor generator tree, function $\mathcal{O}_{out}(a, n)$ in Algorithm 1 computes the set of operators applicable in $a$.*

*Proof sketch.* A Cartesian state $a$ is a Cartesian set of concrete states $S$. We can compute the set of operators applicable in at least one $s \in S$ by looping over $S$, querying the successor generator for $s$ and collecting all reported operators. Algorithm 1 interleaves these traversals by considering all states $s \in S$ at the same time. $\square$

### Incoming Operators

If we remove all mentions of outgoing state $a$ from Proposition 1, we see that the set of operators that can reach an abstract state $b$ is $\mathcal{O}_{in}(b) = \{o \in \mathcal{O} \mid \mathcal{C}(post(o)) \cap b \neq \emptyset\}$. Again, the naive way of computing this set by looping over all operators can be improved upon by using the successor generator data structure. We do so by constructing a successor generator for the set of inverted operators $\mathcal{O}' = \{\langle post(o), pre(o)\rangle \mid o \in \mathcal{O}\}$, i.e., instead of branching over preconditions, we branch over postconditions.

### Outgoing Transitions

Now that we have a method for efficiently computing the set of outgoing operators $\mathcal{O}_{out}(a)$ in a given abstract state $a$, we need to efficiently find the set of abstract states $b$ that can be reached from $a$ via an operator $o \in \mathcal{O}_{out}(a)$. We can formalize this set by $T_{out}(a, o) = \{a \xrightarrow{o} b \mid b \in S(\mathcal{T}'), b \cap progr(a, o) \neq \emptyset\}$. The naive computation of this set loops over all states $b \in S(\mathcal{T}')$ and checks whether $b$ overlaps with $progr(a, o)$. Since each iteration of the refinement loop adds another abstract state, this computation will run slower and slower over time.

To compute $T_{out}(a, o)$ efficiently, we turn to another tree data structure, the *refinement hierarchy*, which holds a record of all refinements (Seipp and Helmert 2018).[2] Each

---

[2]To simplify the presentation, we assume that each refinement splits off a single atom. To account for splitting off multiple atoms, our implementation uses a directed acyclic graph instead of a tree.

**Algorithm 2** Compute the set of abstract states that share at least one concrete state with Cartesian set $c$, starting from refinement hierarchy root node $n$.

1: **function** INTERSECT$(c, n)$
2:    **if** $n$ is leaf **then**
3:       **yield** $n$
4:    **else**
5:       **if** $dom(n.left, n.var) \cap dom(c, n.var) \neq \emptyset$ **then**
6:          **yield from** INTERSECT$(c, n.left)$
7:       **if** $dom(n.right, n.var) \cap dom(c, n.var) \neq \emptyset$ **then**
8:          **yield from** INTERSECT$(c, n.right)$

node in this binary tree represents a Cartesian set and the leaf nodes are the abstract states in the current abstraction. Each non-leaf node $n$ holds the variable $n.var$ for which the associated Cartesian set was split and pointers to the two resulting child nodes $n.left$ and $n.right$.

Algorithm 2 shows the INTERSECT function which uses the refinement hierarchy with root node $n$ to compute the set of abstract states that intersect with a given Cartesian set $c$. We use it to compute $T_{out}(a, o)$ as INTERSECT$(progr(a, o), n)$.

**Proposition 3.** *For Cartesian set $c$ and root node $n$ of a refinement hierarchy for abstraction $\mathcal{T}'$, INTERSECT$(c, n)$ computes the set of abstract states in $\mathcal{T}'$ that overlap with $c$.*

*Proof sketch.* When intersecting two Cartesian sets, we can consider each variable independently of the others. INTERSECT uses this to compute the overlapping states recursively, at each node $n$ checking for which of the children the intersection for the split variable $n.var$ is non-empty. $\square$

Even though we need to follow at least one child node at each internal node, the fact that the depth of the refinement hierarchy is bounded by the number $N$ of atoms in $\Pi$ makes INTERSECT an appealing alternative to looping over all $O(2^N)$ states in the abstraction.

### Incoming Transitions

Similarly to the outgoing transitions, the transitions induced by operator $o$ that lead into state $b$ are $T_{in}(b, o) = \{a \xrightarrow{o} b \mid a \in S(\mathcal{T}'), a \cap regr(b, o) \neq \emptyset\}$. To compute this set efficiently, we call INTERSECT$(regr(b, o), n)$.

### Caching Optimal Transitions

There is a middle ground between storing all transitions and storing no transitions: we can store only *optimal* transitions. A transition $a \xrightarrow{o} b$ is optimal iff $h^*_{\mathcal{T}'}(a) = cost(o) + h^*_{\mathcal{T}'}(b)$, where $h^*_{\mathcal{T}'}(x)$ is the cost of a cheapest path from $x$ to a goal state in $S_\star(\mathcal{T}')$. The CEGAR algorithm uses incremental search (Seipp, von Allmen, and Helmert 2020) to maintain for each state $a$ a transition $a \xrightarrow{o} b$ that starts a cheapest path from $a$. In several places of the algorithm the incremental search only needs access to the optimal transitions, so by caching them, we can often avoid computing *all* transitions.

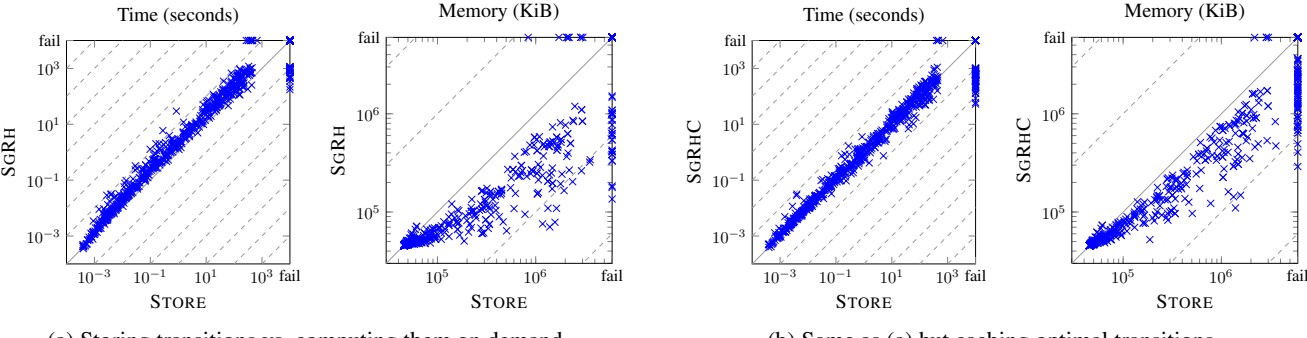

| | Time (seconds) | Memory (KiB) | Time (seconds) | Memory (KiB) |
|---|---|---|---|---|

(a) Storing transitions vs. computing them on demand.  (b) Same as (a) but caching optimal transitions.

Figure 1: Time and peak memory usage for refinement loop executions that find a concrete solution. Runs that exhaust the time or memory limit appear on "fail" axes.

| | | STORE | NAIVE | SG | RH | SGRH | SGRHC |
|---|---|---|---|---|---|---|---|
| refine | solved | 580 | 352 | 353 | 582 | 600 | **637** |
| | out of time | 162 | 1462 | 1461 | 1230 | 1211 | 423 |
| | out of mem. | 1072 | – | – | 2 | 3 | 754 |
| search | solved | 255 | 436 | 437 | 258 | 246 | 211 |
| | out of time | 1 | 21 | 13 | 10 | 8 | 6 |
| | out of mem. | 978 | 1005 | 1011 | 964 | 960 | 960 |
| | solved total | 835 | 788 | 790 | 840 | 846 | **848** |

Table 1: Number of occurences of different outcomes for the refinement loop and the A* search. We count both "solution found" and "proved unsolvable" as solved and omit the 13 tasks for which the translator runs out of memory.

## Experiments

We implemented our algorithms in the Scorpion planning system, which is an extension of Fast Downward (Helmert 2006) and used the Downward Lab toolkit (Seipp et al. 2017) for running experiments. Our benchmark set consists of all 1827 tasks without conditional effects from the optimal sequential tracks of the International Planning Competitions 1998–2018. We limit runtime to 30 minutes and memory to 4 GiB. When the refinement loop exhausts the internal time limit of 20 minutes or tries to use more than 3.5 GiB of memory, we stop refining and use the resulting heuristic in an A* search. All benchmarks, code and experiment data are available online (Reference removed for review).

We compare the previous state of the art (STORE) to five variants of our algorithms in an ablation study. Before inspecting heuristic quality, we evaluate the effects on the refinement loop, which is our main focus. Table 1 shows that if we store all transitions in memory (STORE), we solve 580 tasks during refinement, but run out of memory for the vast majority of the remaining tasks. By computing all operators and transitions naively on demand (NAIVE) we never run out of memory, but the refinement loop slows down drastically, leading to solving only 352 tasks during refinement. Using successor generators for computing operators (SG) incurs up to a ten-fold speedup for some commonly solved tasks, but

this only translates to solving one extra task during refinement (353 tasks in total). In contrast, computing transitions using the refinement hierarchy (RH), while computing operators naively, leads to solving 582 tasks during refinement, a 65% increase over SG. Using the tree data structures for both computations (SGRH) leads to solving 600 tasks during refinement, while still almost never running out of memory. Finally, caching all optimal transitions (SGRHC) hits the sweet spot between memory usage and runtime and solves 637 tasks during refinement, 57 tasks more than STORE.

Figure 1 compares our strongest variants, SGRH and SGRHC, to STORE in terms of runtime and memory consumption during the refinement loop. The plots visualize the time vs. memory trade-off: while SGRH is slightly slower than STORE, it uses much less memory. SGRHC uses more memory than SGRH but still less memory than STORE for most tasks. As a result, SGRH is roughly as fast as STORE.

Regarding heuristic accuracy, Table 1 shows that all algorithm variants suffer from diminishing returns: solving additional tasks during the refinement becomes harder and harder and all variants benefit from switching from the refinement loop to an A* search eventually. We also see that all resulting heuristics are so fast to evaluate that runtime almost never becomes a bottleneck. Our strongest algorithm variants solve more tasks overall (up to 848 tasks) than the previous state of the art (STORE: 835 tasks). This is the case not only since more tasks are solved during refinement, but also since the resulting heuristics are more accurate. SGRHC computes a higher lower bound than STORE for 633 tasks, while the opposite is only true for 152 tasks. Also, SGRHC needs fewer expansions than STORE until the last $f$ layer for 217 tasks, while the opposite only holds for 15 tasks.

## Conclusions

Our algorithms for efficiently computing transitions in Cartesian abstractions drastically reduce the memory usage during the refinement loop, while only slowing it down slightly. If we store all optimal transitions, we can trade a bit of memory for faster runtime and solve even more tasks.

In future work, we want to evaluate whether the benefits of our algorithms for single abstractions carry over to the setting where we compute multiple Cartesian abstractions.

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
