# OpenReview forum: "Efficiently Computing Transitions in Cartesian Abstractions"
_icaps-conference.org/ICAPS/2024/Conference — ICAPS 2024_

### Official Review · Reviewer_tDJD · 2024-01-08

**Significance And Importance:** 2
**Soundness:** 4
**Novelty:** 2
**Clarity:** 4
**Overall Evaluation:** 2
**Confidence:** 4

**Weaknesses:**

2: No major or minor weaknesses.

**Contributions Of The Paper:**

The paper provides an algorithm setup for on-demand computation of abstract transition in counterexample-guided cartesian abstraction refinement.

**Ethical Considerations:**

(5) Excellent: The paper comprehensively addresses all of the applicable ethical considerations

**Nomination For Best Paper:**

No

**Questions For Authors:**

(1) Can you provide some intuition why the heuristics resulting from your method are more accurate (cf., line 307)?

Typos:
296: "SGRH is roughly as fast as STORE" I believe, it should be "SGRHC"?.

**Reproducibility:**

4: Authors promise to release code and domains (whichever apply).

**Strengths Of The Paper:**

The method achieves significant decrease in memory-consumption and an increase in coverage compared to the SOA-version of counterexample-guided cartesian abstraction refinement.
The paper cleverly adopts (successor generator) and reuses (refinement hierarchy) existing data structures.

**Weaknesses Of The Paper:**

The performance gain seems to be mostly in terms of memory "only". The increase in coverage is modest. Overall the contribution is somewhat incremental.

---

> ### Author Rebuttal · Authors · 2024-01-27
>
> > Can you provide some intuition why the heuristics resulting from your method are more accurate (cf., line 307)?
>
> Our new heuristics exhaust the available memory less often, so they can spend more time in the refinement loop. Each refinement can only make the heuristic more accurate.

---

### Official Review · Reviewer_XXYa · 2024-01-23

**Significance And Importance:** 2
**Soundness:** 4
**Novelty:** 2
**Clarity:** 4
**Overall Evaluation:** 1
**Confidence:** 3

**Weaknesses:**

1: Minor weaknesses that are easily fixable.

**Contributions Of The Paper:**

The paper introduces a more memory-friendly method to generate cartesian
abstractions via counterexample-guided abstraction refinement.

The memory bottle-neck of storing the transitions between abstract states
is addressed by re-computing the needed transitions on the fly.

To do so, they build on existing data structures.
To compute which operators are applicable in a cartesian state, the
Fast Downward successor generator is used.
To then compute which abstract states are reachable by those operators they
repurpose the refinement tree, that represents the split of abstract states
during the refinement.

They identify, that recomputing all transitions is too time-consuming and
instead, propose to only recompute non-optimal transitions.

**Ethical Considerations:**

(1) Not Applicable: The paper does not have any ethical considerations to address

**Nomination For Best Paper:**

No

**Questions For Authors:**

I have no further questions for the authors.

**Reproducibility:**

4: Authors promise to release code and domains (whichever apply).

**Strengths Of The Paper:**

The paper is well-written and structured.
They clearly describe the advantages of the proposed approach over the
trivial computation.

They address a common problem in abstract state generations by elegantly
repurposing existing data structures.

The experiments sufficiently evaluate the impact of the individual
components of the transition generation and clearly show the memory decrease
vs only a moderate increase in run time.

They identify the sweet spot for storing vs generating transitions by
only storing the optimal transitions.

This strategy results in an increase in solved instances during the
refinement of the abstraction as well as to a more informed heuristic.

**Weaknesses Of The Paper:**

The paper presents only an incremental advance.
The number of instances solved during refinement increases significantly,
However, the overall number of solved tasks increases only slightly.

Small notes:

How dom(v_i, a) is introduced in line 117 is a little bit confusing
because "a" is only later specified as a = A_1 \times...

The use of yield in the pseudo-code was first confusing and made it harder
to understand.

---

> ### Author Rebuttal · Authors · 2024-01-27
>
> Thank you for your review! We will take your suggestions into account for making the presentation clearer.
>
> > However, the overall number of solved tasks increases only slightly.
>
> This is somewhat to be expected since task difficulty scales exponentially with task size in classical planning. In any case, our work focuses more on the refinement loop and less on the search algorithm.

---

### Official Review · Reviewer_9Scg · 2024-01-23

**Significance And Importance:** 3
**Soundness:** 4
**Novelty:** 3
**Clarity:** 4
**Overall Evaluation:** 2
**Confidence:** 2

**Weaknesses:**

2: No major or minor weaknesses.

**Contributions Of The Paper:**

- A memory efficient solution by computing transition abstractions on demand.
- Increased number of solved tasks in comparison to the state of the art, with fast and accurate heuristic estimation.

**Ethical Considerations:**

(1) Not Applicable: The paper does not have any ethical considerations to address

**Nomination For Best Paper:**

No

**Questions For Authors:**

- The paper is very well written and with detailed explanations. I do not have major questions or issues with the paper. My familiarity with abstractions for computing heuristics is low.

- Is there any scenario (say worst case in complexity) that the proposed solution might perform worse?

**Reproducibility:**

4: Authors promise to release code and domains (whichever apply).

**Strengths Of The Paper:**

- Memory efficient with slight computation overhead.
- Computing heuristic values does not incur much overhead.
- The authors take advantage of efficient data structures, i.e. the successor generator from Helmert 2006.

**Weaknesses Of The Paper:**

- Although memory efficient, the proposed solution trades a bit on computation overhead. It would be interesting to have some discussion on the complexity of the solution.

---

> ### Author Rebuttal · Authors · 2024-01-27
>
> Thank you for your review!
>
> > Is there any scenario (say worst case in complexity) that the proposed solution might perform worse?
>
> The worst-case runtime of the successor generator is linear in the number of operators (like the naive method), but in the best case it is linear in the number of *applicable* operators. We hypothesize that the worst-case runtime complexity for the Intersect function is linear in the number of abstract states, just like its naive counterpart.

---

### Meta-Review · Area_Chair_FQBA · 2024-02-01

**Recommendation:** Accept (Poster)
**Confidence:** 5

**Metareview:**

Clear case, congratulations.

I recommend poster acceptance here as the topic seems rather specialized and of interest to a small subset of the community.

**Ethical Considerations:**

(4) Good: The paper adequately addresses most, but not all, of the applicable ethical considerations